# Retrospective cohort study to evaluate medication use in patients hospitalised with COVID-19 in Scotland: protocol for a national observational study

Tanja Mueller [ID] ,[1,2] Steven Kerr,[3] Stuart McTaggart [ID] ,[4] Amanj Kurdi [ID] ,[1,2] Eleftheria Vasileiou [ID] ,[3] Annemarie Docherty,[3] Kenny Fraser,[5] Ting Shi [ID] ,[3] Colin R Simpson,[3,6] Marion Bennie [ID] ,[4,7] Aziz Sheikh[3,8]

For numbered affiliations see end of article.

**Correspondence to**
Dr Tanja Mueller;
tanja.muller@strath.ac.uk

## ABSTRACT

**Introduction** COVID-19 has caused millions of hospitalisations and deaths globally. A range of vaccines have been developed and are being deployed at scale in the UK to prevent SARS-CoV-2 infection, which have reduced risk of infection and severe COVID-19 outcomes. Those with COVID-19 are now being treated with several repurposed drugs based on evidence emerging from recent clinical trials. However, there is currently limited real-world data available related to the use of these drugs in routine clinical practice. The purpose of this study is to address the prevailing knowledge gaps regarding the use of dexamethasone, remdesivir and tocilizumab by conducting an exploratory drug utilisation study, aimed at providing in-depth descriptions of patients receiving these drugs as well as the treatment patterns observed in Scotland.

**Methods and analysis** Retrospective cohort study, comprising adult patients admitted to hospital with confirmed or suspected COVID-19 across five Scottish Health Boards using data from in-hospital ePrescribing linked to the Early Estimation of Vaccine and Anti-Viral Effectiveness (EAVE II) COVID-19 surveillance platform. The primary outcome will be exposure to the medicines of interest (dexamethasone, remdesivir, tocilizumab), either alone or in combination; exposure will be described in terms of drug(s) of choice; prescribed and administered dose; treatment duration; and any changes in treatment, for example, dose escalation and/or switching to an alternative drug. Analyses will primarily be descriptive in nature.

**Ethics and dissemination** Ethical and information governance approvals have been obtained by the National Research Ethics Service Committee, South East Scotland 02 and the Public Benefit and Privacy Panel for Health and Social Care, respectively. Findings from this study will be presented at academic and clinical conferences, and to the funders and other interested parties as appropriate; study findings will also be published in peer-reviewed journals. Publications will be available on the EAVE II website (https://www.ed.ac.uk/usher/eave-ii/key-outputs/our-publications), alongside lay summaries and infographics aimed at the general public. Press releases will also be considered, if appropriate.

## Strengths and limitations of this study

► This study will use data collected as part of routine care to address prevailing knowledge gaps with regard to the treatment of hospitalised COVID-19 patients.
► In-patient electronic prescribing data will be linked with a wide range of other datasets, enabling an in-depth description of current clinical practice in Scotland.
► Analyses will mainly be descriptive in nature; although comprising basic testing for associations between variables, causal analyses will be outwith the scope of this study due to its observational nature.

## INTRODUCTION

Since first appearing in Wuhan, China, in late 2019, the new severe acute respiratory syndrome coronavirus 2 (SARS-CoV-2) has spread globally, resulting in the WHO first declaring a Public Health Emergency of International Concern and then, in March 2020, a pandemic.[1] The disease caused by SARS-CoV-2 is now widely known as COVID-19.

Early symptoms of COVID-19 tend to occur between 5 and 10 days after infection, and commonly include fever, loss of smell and/or taste, and a persistent cough.[2] Symptoms may become increasingly severe over a period of approximately 2 weeks, and can lead to hospitalisation mainly due to breathing problems; patients with severe disease frequently require mechanical ventilation.[3] COVID-19 potentially leads to organ damage, and can result in long-term health problems ('long COVID-19').[4] Disease outcomes generally appear to be linked to age and pre-existing conditions, including cardiovascular diseases and diabetes.[5]

A number of COVID-19 vaccines have been developed and are now being successfully

deployed at scale in the UK.[6 7] Furthermore, while the condition itself is self-limiting in the majority of cases, a range of repurposed drugs are currently being used to alleviate symptoms and/or decrease mortality in hospitalised COVID-19 patients, mostly based on evidence emerging from clinical trials—including, for example, antiviral drugs that have previously been tested in conditions caused by similar viruses such as SARS or Middle East respiratory syndrome (MERS) or other viral infections such as HIV or Ebola[8 9]; anti-inflammatory drugs including corticosteroids[10] and monoclonal antibodies[11–14]; and a raft of other drugs, from antibiotics[15] to interferons.[16] In addition, convalescent plasma therapy has been proposed.[17]

With interest in this area remaining high, new study results being reported on a frequent basis, and several clinical trials still ongoing, treatment recommendations are rapidly updated[18]; therefore, treatment guidelines—and, consequently, clinical practice—are likely to differ substantially, both across countries and over time. For instance, a recent multinational cohort study has investigated the use of repurposed and adjuvant drugs in hospitalised COVID-19 patients in China, South Korea, Spain, and the USA, and found that azithromycin, the antivirals lopinavir and ritonavir, and the antimalaria drug hydroxychloroquine were frequently used at the beginning of the pandemic; however, following reports of the non-effectiveness of these drugs in combination with safety issues related to hydroxychloroquine, their use has declined, and dexamethasone and remdesivir use have instead been increasing. In addition, use patterns differed considerably between these countries.[19]

Dexamethasone,[20] remdesivir[21] and tocilizumab[22] have been recommended for use in hospitalised patients with severe COVID-19 within the UK based on randomised controlled trial evidence, most prominently the 'Randomised Evaluation of COVID-19 Therapy' (RECOVERY) trial.[23] There is, however, currently limited real-world evidence available related to the use of these drugs in routine clinical practice. For instance, it is thus far unclear which patients are being prescribed dexamethasone, remdesivir and/or tocilizumab as part of their in-hospital treatment, and at what point; what the most common treatment patterns are; how the use of these drugs has changed since the start of the pandemic; and whether there are any geographical differences observable. Further evidence on the real-world clinical effectiveness and safety of these drugs is also required.[24]

There have been few published studies of in-hospital drug utilisation. This has been due, in part, to patient-level data being unavailable as drug prescribing and administration records are paper-based in many secondary care settings.[25] The implementation of electronic prescribing in hospitals in Scotland has simplified data sharing across healthcare settings. The wider roll-out of the 'Hospital Electronic Prescribing and Medicines Administration' (HEPMA) system was initiated in 2014[26] in line with the Scottish eHealth strategy,[27] and HEPMA is now available

to hospitals across five out of the 14 Health Boards in Scotland (regional organisations responsible for delivering healthcare to their respective populations).[28]

Our aim is to contribute to addressing the prevailing knowledge gaps by conducting an exploratory drug utilisation study using data from in-hospital ePrescribing linked to the Early Estimation of Vaccine and Anti-Viral Effectiveness (EAVE II) COVID-19 surveillance platform.[24] This linked data will be analysed to provide an in-depth description of the treatments hospitalised COVID-19 patients receive (whether alone or in combination), and to describe the outcomes in these patients. The drugs of interest include dexamethasone, remdesivir and tocilizumab, based on information requests by clinicians working in this setting; these drugs are currently routinely used in patients hospitalised with COVID-19 in Scotland due to recent treatment recommendations.

### Primary objectives

► Identify patients being treated with dexamethasone, remdesivir and/or tocilizumab (either as monotherapy or in combination) for COVID-19 after being admitted to hospital as part of standard care.
► Describe and summarise baseline characteristics of these patients, including COVID-19 status (suspected at hospital admission based on symptoms, vs confirmed via PCR test); sociodemographics (age; sex; health board; deprivation; hospital type; admission from care home; hospital readmission); and clinical variables potentially related to treatment choice and/or possible outcomes (including, but not restricted to, comorbidities,[29] concomitant medication and intensive care unit (ICU) admission).
► Describe treatment patterns, including drug chosen and dose administered; treatment duration; dose escalations; and changes in the drug given (eg, switching from dexamethasone to hydrocortisone or methylprednisolone).
► Describe patterns of medicines use over time and across geographical areas, and potentially by patient characteristic as and when appropriate.
► Map out patient pathways and describe admission episodes and their outcomes (hospital admission details and duration of in-hospital stay, ICU transferal, administration of dexamethasone or any other drug of interest, discharge or death); potentially stratified by patient characteristics as and when appropriate.

### Secondary objective

► Evaluate the impact of guideline changes on the patterns of use of dexamethasone, remdesivir and tocilizumab over time.

### METHODS AND ANALYSIS
### Study design

Retrospective cohort study, comprising adult patients (18 years of age or older) admitted to hospital with confirmed or suspected COVID-19 across five Scottish Health

Boards: NHS Ayrshire and Arran, NHS Dumfries and Galloway, NHS Forth Valley, NHS Lanarkshire and NHS Lothian. The total population size of these health boards was approximately 2.4 m people (~45% of the Scottish population) in mid-2019.[30] However, since implementation of HEPMA in NHS Lothian happened later than in the other Health Boards, data might not be as complete, particularly for the early months of the study period.

## Data sources

The data to be used for this study is part of the EAVE II platform, which has been implemented to determine COVID-19 related risk factors and the COVID-19 healthcare burden; and to evaluate the uptake, safety and effectiveness of therapeutic interventions.[24] All data have been collected as part of routine care.

The EAVE II data source contains primary healthcare records, linked with patient-level secondary care data using the Community Health Index number[31]—a unique patient identifier used throughout the Scottish health system—and comprises the following datasets:

► COVID-19 test results: Electronic Communication of Surveillance Scotland (ECOSS).[32]
► Vaccination status: Turas Vaccine Management Tool,[33] General Practice (GP) extract.
► Hospital admissions and in-patient episodes: Scottish Morbidity Record (SMR01), Rapid Preliminary Inpatient Data (RAPID) and Scottish Intensive Care Society Audit Group database.[34]
► In-hospital medicines use and community prescriptions: HEPMA[35] and Prescribing Information System (PIS),[34] respectively.
► Mortality: National Records of Scotland.[34]

HEPMA will be used for identification of the main study outcomes, including medication use patterns; all available datasets may be used to identify other study outcomes as appropriate and feasible.

EAVE II data are held by Public Health Scotland; pseudonymised data will be accessed using the National Safe Haven, a secure, closed environment.[36]

## Study population

The study population will comprise all adult patients admitted to hospital in the five aforementioned health boards since 1 March 2020 with a primary diagnosis of COVID-19, up to the latest date available. COVID-19 hospitalisations will be defined as hospitalisations within 28 days of a positive PCR test, or based on an admission with an ICD-10 code for COVID-19 (U07.1 and U07.2) as recorded in hospital episode records (SMR01 and/or RAPID); ICD-10 diagnoses will be confirmed using available PCR test results (ECOSS) where possible. Hence, the population will include both laboratory confirmed and clinical based diagnoses, respectively.

Patients will be followed up from the index date, defined as the first prescribing date for any of the medications of interest (the exposure), until discharge from

hospital, death, or the end of the study period subject to data availability, whichever occurs first.

Patients receiving any of the drugs as part of a clinical trial will be excluded for analyses based on trial participation information if available (eg, trial flag in HEPMA); or based on the dates where drugs became recommended for use in daily practice as communicated by the Scottish Government/NHS Scotland (see also table 1 below for details).

## Primary outcome

The primary outcome will be treatment with the medicines of interest, either alone or in combination, with a particular emphasis on dexamethasone as dexamethasone is the most widely used of these drugs, and the availability of both prescribing and administration data is expected to be high. All patients receiving dexamethasone will be included in the first instance; however, analyses will mainly focus on those receiving the recommended dosing regimen for patients with COVID-10 (6 mg orally or 1.8 mL intravenously, once daily for 10 days)[20] since other dosing regimens are more likely being prescribed for indications other than COVID-19. Alternative recommended corticosteroids such as prednisolone and hydrocortisone will also be considered (recommended doses: 40 mg orally once daily for 10 days; 50 mg intravenously every 8 hours for 10 days, respectively). Remdesivir and tocilizumab will be included for analyses where sufficient data are available; since these two drugs are given intravenously, the data available might be limited (ie, exclude the exact dose administered).

Exposure will be described in terms of drug(s) of choice; prescribed and administered dose; treatment duration; and any changes in treatment, for example, dose escalation and/or switching to an alternative drug (including time to dose escalation/switching and reasons for these, if available). The first drug prescribed on HEPMA following admission to hospital on or after the date of (possible) COVID-19 diagnosis will be defined as the index drug (ie, dexamethasone, remdesivir or tocilizumab); the date of the first recorded prescription will be used as the index date (for the purpose of setting the baseline). Duration of treatment will be calculated using the dates of first/last recorded administration of the drug in question.

## Other outcomes

Additional outcomes relating to the primary study objective include hospital specialty at admission, in-hospital transfer (eg, admission to ICU), length of stay, and outcome of hospital episode (discharge or death).

Secondary outcomes include in-hospital mortality, that is, death on the same day as discharge; and out-of-hospital mortality following discharge, if feasible.

## Covariates

Patient characteristics of interest that might potentially influence choice of drug, duration of treatment and

**Table 1** Description of variables (cohort identification, outcomes, covariates)

| Variable | Data source | Description | Value |
|---|---|---|---|
| **Cohort: COVID-19 status** | | | |
| Cause of admission | SMR01/RAPID | Suspected or confirmed | ICD10 codes: U07.1, U07.2 |
| PCR test result | ECOSS | COVID-19 test result (within 28 days prior to admission) | Categorical: positive, negative, unavailable |
| **Primary outcome: medication use** | | | |
| Drug name | HEPMA | Drugs of interest: Dexamethasone, remdesivir, tocilizumab* | Character (name, according to dm +d) |
| Drug dose | HEPMA | Prescribed and administered | Numeric (mg, ml) |
| Prescribed date | HEPMA | First prescribed date: index date | Date (yyyy-mm-dd) |
| Administered date | HEPMA | Dates of drug administration | Date (yyyy-mm-dd) |
| Duration of treatment† | HEPMA | (first—last administered date)‡ | Numeric (days) |
| Treatment changes† | HEPMA | Changes in dosing and/or drug | Categorical: yes, no |
| **Secondary outcomes** | | | |
| Hospital specialty | SMR01 | At admission | Character (name) |
| Specialty changes† | SMR01 | Internal transferals during stay | Categorical: yes, no |
| ICU/HDU | SICSAG | Admission to intensive care | Categorical: yes, no |
| Discharge: alive | SMR01 | Outcome of hospital episode | Categorical: home, w/family, care facility, other hospital |
| Discharge: dead | SMR01, NRS | Outcome of hospital admission (In-hospital mortality) | Categorical: yes, no Cause: ICD-10 codes |
| Death | NRS | Overall mortality (after discharge) | Categorical: yes, no Cause: ICD-10 codes |
| Length of stay§ | SMR01 | Duration of in-hospital stay | Numeric (days) |
| **Covariates: sociodemographic** | | | |
| Age† | GP extract | Patient age at index date | Numeric (years) |
| Sex | GP extract | Biological sex at birth | Categorical: male, female |
| Health board | GP extract | Patient place of residence at admission | Categorical: A&A, D&G, Forth Valley, Lanarkshire, Lothian; |
| Data zone | GP extract | Patient place of residence | Categorical |
| SIMD | GP extract | Level of deprivation, based on data zone of residence | Categorical: 1 (most) to 5 (least deprived) |
| **Covariates: disease (severity) related** | | | |
| COVID-19 vaccination status | GP extract/ TVMT | Status at hospital admission | Categorical: unvaccinated, vaccinated once, twice |
| Level of care | SICSAG/ SMR01 | Admission to ICU; level of care received while at ICU | Categorical: yes, no Categorical: ACP levels 0–3 |
| Supporting medication | HEPMA | Therapeutics prescribed and administered during in-hospital stay | Character (name, according to dm +d) |
| **Covariates: comorbidities and concomitant medication** | | | |
| Other causes of admission | SMR01 | Conditions underlying or attributing to hospital admission | ICD-10 codes |
| Comorbidities | GP extract/ SMR01 | Pre-existing conditions | READ codes, ICD-10 codes |
| Charlson score†[41] | SMR01 | Estimated based on secondary care data (historic hospital episodes) | Numeric |
| Concomitant medication | PIS | Potential proxy for comorbidities; specific drugs of interest | Character (name, according to the BNF) |

Continued

| Table 1 | Continued | | |
|---------|-----------|---|---|
| Variable | Data source | Description | Value |
| Polypharmacy † | PIS | Based on number of different drugs prescribed simultaneously | Categorical: yes, no |

*Cut-off dates to exclude patients who have been treated as part of a clinical trial, if no trial flag participation available in the dataset: remdesivir 29 May 2020; dexamethasone 16 June 2020; tocilizumab 8 January 2021.

†Denotes derived variables.

‡Adding discharge/outpatient prescribing if patient discharged prior to end of treatment regimen (if available).

§Can be derived if variable not readily available in dataset (date of discharge—first date of admission).

A&A, Ayrshire & Arran; ACP, augmented care period; BNF, British National Formulary; D&G, Dumfries & Galloway; dm+d, dictionary of medicines and devices; ECOSS, Electronic Communication of Surveillance in Scotland; GP, General Practice; HDU, high-dependency unit; HEPMA, Hospital Electronic Prescribing and Medicines Administration; ICD-10, International Classification of Diseases, 10th Edition; ICU, intensive care unit; NRS, National Records of Scotland; PIS, prescribing information system; RAPID, rapid preliminary in-patient data; SICSAG, Scottish Intensive Care Society Audit Group; SIMD, Scottish Index of Multiple Deprivation; SMR01, Scottish Morbidity Records, in-patient dataset; TVMT, Tuas Vaccine Management Tool.

(possibly) treatment outcomes will be identified and summarised at baseline, and comprise sociodemographic factors (age, sex, health board of residence, level of deprivation); disease-related aspects; comorbidities; and concomitant medication.

The level of deprivation will be characterised using the Scottish Index of Multiple Deprivation, an area index combining information with regards to health, access to services, education, employment, income, housing and crime.[37] Disease-related aspects refer to information potentially linked to disease severity, for example, level of hospital care/additional treatments received (ICU admission, mechanical ventilation) if and where available; while $O_2$ saturation levels would be highly relevant, particularly with regard to treatment outcomes, this information is not present in the available dataset. Comorbidities of interest will comprise mainly those conditions used to identify patients at high risk of adverse outcomes (ie, shielding list),[29] for example, respiratory disease (asthma, chronic obstructive pulmonary disease), cardiovascular diseases, diabetes (type 1 and type 2), chronic kidney disease and cancer; other comorbidities might also be included. Concomitant medication at baseline will focus on drugs potentially impacting the immune system and/or affecting the risk of infections (immunosuppressants, steroids, antimicrobial drugs), and those with an (hypothesised) effect on disease severity or outcome—either directly or as a proxy for underlying conditions potentially not captured otherwise within the dataset (eg, long-acting muscarinic antagonists/long acting beta-agonists, insulin and antidiabetic drugs, anticoagulants, antiplatelet drugs, ACE inhibitors, angiotensin receptor blockers). If possible, COVID-19 vaccination status of patients will also be assessed. Furthermore, the presence of polypharmacy—defined as the simultaneous use of five or more different medications prior to being admitted to hospital—will be identified.

Baseline characteristics will be defined using all available data, with restrictions on included time periods (mainly with regard to concomitant medication) based on the specifications used in previous studies.[5 6 24]

Exposure, outcome, and relevant covariates—alongside the data source, their coding and a brief description—are presented in table 1.

## Statistical analysis

All analyses relating to the primary objectives of this study will be descriptive in nature, and may include counts/frequencies for categorical variables and mean/SD or median/IQR for continuous variables, as appropriate. In addition, patient pathways will be visualised using Sankey plots or similar techniques.

The impact of changes in treatment guidelines on the use of dexamethasone will be evaluated using interrupted time series analysis; logistic regression or time-to-event analysis (eg, Kaplan-Meier plots) will be used to assess discharge patterns or patient mortality, if feasible.

All analyses will be conducted using R/RStudio, V.3.6.1.[38 39]

## Patient and public involvement

The EAVE II Public Advisory group are a diverse group of patient and public involvement (PPI) contributors who meet monthly to incorporate the views of patients and the public into research using the EAVE II dataset. This includes shaping of research via the EAVE II Steering Group, which is attended by our two lay leads. The lay summary for this research will be cowritten with our PPI contributors and shared via the outputs section of the EAVE II website,[36] hosted by the University of Edinburgh.

## ETHICS AND DISSEMINATION

Ethical and information governance approvals have been obtained by the National Research Ethics Service Committee (REC), South East Scotland 02 (REC number: 12/SS/0201) and the Public Benefit and Privacy Panel for Health and Social Care (reference number:

1920-0279), respectively. Findings from this study will be presented at academic and clinical conferences, and to the funders and other interested parties as appropriate. Study findings will also be published in peer-reviewed journals; reporting of findings will follow the STROBE (Strengthening the Reporting of Observational Studies in Epidemiology)[40] and RECORD (Reporting of Studies conducted using Observational Routinely collected Data) guidelines.

**Author affiliations**

[1]Strathclyde Institute of Pharmacy and Biomedical Sciences, University of Strathclyde, Glasgow, UK

[2]Public Health Scotland Glasgow Office, Glasgow, UK

[3]Usher Institute, The University of Edinburgh, Edinburgh, UK

[4]Public Health Scotland, Edinburgh, UK

[5]Triscribe Limited, Glasgow, UK

[6]School of Health, Wellington Faculty of Health, Victoria University of Wellington, Wellington, New Zealand

[7]Institute of Pharmacy and Biomedical Sciences, University of Strathclyde, Glasgow, UK

[8]BREATHE Hub, HDR UK, Edinburgh, UK

**Acknowledgements** We thank Dave Kelly from Albasoft for his support with making primary care data available; and Wendy Inglis-Humphrey and Vicky Hammersley for their support with project management and administration. This work is only possible because of the wealth of information collected by the NHS as part of routine clinical practice.

**Contributors** TM, SK, SM, AK and MB conceptualised the study. EV, AD, KF, TS and CRS provided additional methodological and/or clinical advice. AS is the principal investigator of the EAVE II project and provides strategic advice. TM drafted the protocol. All authors read, critically revised and approved the final draft.

**Funding** This analysis is part of the Early Assessment of COVID-19 epidemiology and Vaccine/anti-viral Effectiveness (EAVE II) study. EAVE II is funded by the Medical Research Council (MC_PC_19075) with the support of BREATHE-The Health Data Research Hub for Respiratory Health (MC_PC_19004), which is funded through the UK Research and Innovation Industrial Strategy Challenge Fund and delivered through Health Data Research UK. Additional support has been provided through the Scottish Government DG Health and Social Care.

**Competing interests** AS reports grants from NIHR, grants from MRC, grants from HDR UK, during the conduct of the study. CRS reports funding from NIHR (UK), MRC (UK), CSO (UK), Health Research Council (NZ) and Ministry for Business, Innovation and Employment (NZ) during the conduct of this study. KF is Director of Triscribe, a company providing data quality services and software support. All other authors report no conflicts of interest.

**Patient and public involvement** Patients and/or the public were involved in the design, or conduct, or reporting, or dissemination plans of this research. Refer to the Methods section for further details.

**Patient consent for publication** Not applicable.

**Provenance and peer review** Not commissioned; externally peer reviewed.

**ORCID iDs**

Tanja Mueller http://orcid.org/0000-0002-0418-4789

Stuart McTaggart http://orcid.org/0000-0001-6060-9019

Amanj Kurdi http://orcid.org/0000-0001-5036-1988

Eleftheria Vasileiou http://orcid.org/0000-0001-6850-7578

Ting Shi http://orcid.org/0000-0002-4101-4535

Marion Bennie http://orcid.org/0000-0002-4046-629X

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
