## [Reviewer comments · BMJ Open]

ARTICLE DETAILS

TITLE (PROVISIONAL)	Retrospective cohort study to evaluate medication use in patients hospitalised with COVID-19 in Scotland: protocol for a national observational study
AUTHORS	Mueller, Tanja; Kerr, Steven; McTaggart, Stuart; Kurdi, Amanj; Vasileiou, Eleftheria; Docherty, Annemarie; Fraser, Kenny; Shi, Ting; Simpson, Colin; Bennie, Marion; Sheikh, Aziz

VERSION 1 – REVIEW

REVIEWER	Garfield, Sara University College London, School of Pharmacy
REVIEW RETURNED	05-Aug-2021

GENERAL COMMENTS	This is a very interesting and highly relevant study. The protocol is well written. I have only two comments and these concern the abstract: 1. Please include your plans for dissemination to the public in the dissemination section. 2. While you are not able to establish causality, your analysis is not entirely descriptive as it includes inferential testing. You are using relational analysis as you are testing for associations between variables. Please updated your statement in the limitations section of the abstract to reflect this.
--

REVIEWER	Lukito , Antonia Anna Pelita Harapan University, Cardiovascular
REVIEW RETURNED	21-Aug-2021

GENERAL COMMENTS	This is an important study protocol, to answer whether dexamethasone, remdesivir and/or tocilizumab really have a beneficial impact on the outcomes of COVID-19 patients in the real world setting.
---

VERSION 1 – AUTHOR RESPONSE

Reviewer: 1

Dr. Sara Garfield, University College London

Comments to the Author:

This is a very interesting and highly relevant study. The protocol is well written. I have only two comments and these concern the abstract:

1. Please include your plans for dissemination to the public in the dissemination section.

To address this oversight, we have added the following sentences to the dissemination section of the abstract: "Publications will be available on the EAVE II website (<https://www.ed.ac.uk/usher/eave-ii/key-outputs/our-publications>), alongside lay summaries and infographics aimed at the general public. Press releases will also be considered, if appropriate."

2. While you are not able to establish causality, your analysis is not entirely descriptive as it includes inferential testing. You are using relational analysis as you are testing for associations between variables. Please updated your statement in the limitations section of the abstract to reflect this.

Thank you for highlighting this. We have updated the limitations statement as follows: "Analyses will mainly be descriptive in nature; although comprising basic testing for associations between variables, causal analyses will be outwith the scope of this study due to its observational nature."